# Systemic Mobilization of Breast Cancer Resistance Protein in Response to Oncogenic Stress

**DOI:** 10.3390/cancers14020313

**Published:** 2022-01-09

**Authors:** Małgorzata Szczygieł, Marcin Markiewicz, Milena Julia Szafraniec, Agnieszka Hojda, Leszek Fiedor, Krystyna Urbanska

**Affiliations:** 1Faculty of Biochemistry, Biophysics and Biotechnology, Jagiellonian University, Gronostajowa 7, 30-387 Kraków, Poland; marcin.revan.markiewicz@gmail.com (M.M.); milena.szafraniec@protonmail.com (M.J.S.); hojdaaga@gmail.com (A.H.); krystyna.urbanska@uj.edu.pl (K.U.); 2Łukasiewicz Research Network—PORT Polish Center for Technology Development, Stabłowicka 147, 54-066 Wrocław, Poland

**Keywords:** ABGD2 upregulation, cancer multidrug resistance, xenobiotic transport, drug efflux, cytokines

## Abstract

**Simple Summary:**

The drug efflux mediated by xenobiotic transporters is one of the best recognized mechanisms of multidrug resistance in cancer that leads to the failure of therapeutic approaches. The aim of our research was to examine the influence of a growing tumor on the activity of xenobiotic transport in the host. Our study reveals a strong correlation between the development of melanoma tumor in mice and the level of breast cancer resistance protein, one of the major xenobiotic transporters, and its transcript in the normal tissues of the hosts distant from the tumor site. The systemic effects of the tumor are confirmed by a drastically enhanced xenobiotic transport, which is correlated with changes in the level of cytokines in blood. Such an unexpected type of tumor–host interaction, which leads to the systemic upregulation of breast cancer resistance protein, and very likely of other xenobiotic transporters too, has broad implications for cancer therapies, including chemotherapy and photodynamic therapy. Our findings shed new light on the biology of cancer and the complexity of cancer–host interactions that should be taken into account in the design of new generations of anti-cancer drugs and personalized medicine.

**Abstract:**

The breast cancer resistance protein (BCRP or ABCG2) involved in cancer multidrug resistance (MDR), transports many hydrophobic compounds, including a number of anti-cancer drugs. Our comprehensive study using a mouse model reveals that a subcutaneously growing tumor strongly affects the expression of BCRP in the host’s normal organs on both the transcriptional and translational level. Additionally, the efflux of BCRP substrates is markedly enhanced. The levels of BCRP and its transcript in normal tissues distant from the tumor site correlate with tumor growth and the levels of cytokines in the peripheral blood. Thus, oncogenic stress causes transient systemic upregulation of BCRP in the host’s normal tissues and organs, which is possibly mediated via cytokines. Because BCRP upregulation takes place in many organs as early as the initial stages of tumor development, it reveals a most basic mechanism that may be responsible for the induction of primary MDR. We hypothesize that such effects are not tumor-specific responses, but rather constitute a more universal defense strategy. The xenobiotic transporters are systemically mobilized due to various stresses, seemingly in a pre-emptive manner so that the body can be quickly and efficiently detoxified. Our findings shed new light on the biology of cancer and on the complexity of cancer–host interactions and are highly relevant to cancer therapies as well as to the design of new generations of therapeutics and personalized medicine.

## 1. Introduction

The homeostasis of cells and organisms largely relies on a properly functioning system of xenobiotic transporters. The enhanced activity of these ATP-dependent enzymes not only affects this balance, but in the case of human diseases may also lead to the failure of therapeutic approaches. Indeed, the drug efflux mediated by xenobiotic transporters is one of the best recognized mechanisms of multidrug resistance in cancer (MDR). In fact, as many as 80% of anti-cancer chemotherapies are ineffective due to this phenomenon, which thus poses one of the major concerns in medicine [1,2,3]. The multidrug-resistant phenotype and cross-resistance in tumors are associated with the simultaneous overexpression of various ABC transporters, mainly P-glycoprotein, multidrug resistance protein 1, and breast cancer resistance protein (BCRP or ABCG2, i.e., ATP-binding cassette protein, subfamily G, member 2) [4]. There are two general categories of MDR, primary (intrinsic) and acquired, and the mechanisms responsible for each type are still unclear [5,6]. The former type of MDR exists prior to therapeutic treatment while the latter develops in response to therapeutic pressure [7]. Primary resistance is related to the fact that tumors are heterogeneous and contain various drug-resistant clones which differ in drug sensitivity. Acquired resistance is associated with an increasing insensitivity to the drugs used in therapy; it may result from mutations associated with target genes or proteins, as well as from alteration in the composition of the tumor microenvironment. Both types of mechanisms can coexist with each other during tumor progression [3,5,7]. Although primary resistance has been found in 50% of cancer patients, the mechanisms of its origin remain unknown. Primarily, this is because the vast majority of studies in humans focus on acquired MDR in cancer cells/tissue, whereas studies on whole organisms and normal tissues and organs are sparse [2,7]. Obviously, due to problematic screening, it is practically impossible to collect representative tissue samples at the very early stages of tumor/cancer development.

BCRP performs multiple functions and accepts a broad spectrum of substrates [8], and is considered to be a key component of self-defense systems in organisms. Its major role is to recognize and actively transport natural and synthetic hydrophobic compounds, including chlorophyll (Chl) derivatives of dietary origin and many anti-cancer drugs [8,9]. We have recently shown that the substrate recognition by BCRP is in part controlled via the substrate–albumin–BCRP interplay [10]. The transporter is expressed at very high levels in the placenta and protects the developing fetus from endo- and exogenous toxins, and is found at the blood–brain barrier, where it protects the brain against harmful compounds. It also regulates the homeostasis of nutrients and certain hormones. In the gastrointestinal tract the transporter plays a role in nutrient absorption, helps to concentrate vitamins and possibly also hormones in breast milk, and may regulate testosterone levels in the prostate, where it is expressed in normal prostate basal epithelial cells. The sebaceous glands, exocrine glands located in the skin that secrete sebum to lubricate and water-proof skin and hair, also express a high level of BCRP [1,11,12].

The overexpression of BCRP in tumor tissue is responsible for drug resistance in leukemia and several solid tumors originating in the digestive tract, endometrium, and lung [13], but not in melanoma [14]. It diminishes the efficacy of traditional chemotherapy, e.g., with mitoxantrone, doxorubicin, topotecan, or flavopiridol, as well as that of photodynamic treatment based on Chl derivatives, such as pheophorbide a (Pheide), chlorin e5, or precursor 5-ALA, and therapies based on tyrosine kinase inhibitors (Imatinib, Gefitinib). It is not entirely clear how the upregulation of BCRP is achieved because the control of BCRP expression is a very complex and incompletely understood process, regulated via gene amplification, and at both the transcriptional and translational levels [12,15]. The BCRP regulation pathways may involve nuclear steroid receptors, which act as transcription factors [16,17]. Locally, i.e., within single organs or tissues, the expression of BCRP is under the control of several transcriptional factors, including NF-κB, HIF-1α, PPAR-γ, and Nrf2 [12], and post-translational factors [18] and epigenetic factors, such as CpG island (de)methylation and histone deacetylation [15].

Alterations in the expression of ABC transporters can be observed in many diseases related to inflammation [19], hypoxic diseases [20], and cancer [17], and similar effects have been seen in arthritic rats [21]. Therefore, the involvement of systemic inflammation factors in such a regulation seems quite likely. Indeed, pro-inflammatory cytokines secreted during inflammation seem to alter the expression and activity of proteins from the ABC family [19,22], at both the transcriptional and the post-translational level [23]. Recent clinical studies show that tumor-associated inflammatory response may contribute to the considerable variability in cytotoxic drug clearance in cancer patients [24]. Most of these studies, however, were carried out either on the cellular level or focused on tumor tissue, and in some cases on single organs. Furthermore, the upregulation of xenobiotic transporters in normal tissues has only been reported rarely, whereas in most cases the downregulation of other ABC-family transporters in distant and non-malignant tissues in the host has been observed [20,25]. In this context, the question remains open as to whether and how tumors systemically affect the host’s xenobiotic transport system. These open questions prompted us to investigate the impact of the developing tumor on the expression of BCRP in normal tissues and the efficiency of xenobiotic transport in the host in an animal model. In order to minimize interference from other factors, we chose S91 Cloudman mouse melanoma as the model tumor because it is known to be non-metastatic and shows an intrinsically low expression of BCRP. The expression of BCRP was monitored on both the protein and the transcript levels in normal host tissues that are known to be involved in xenobiotic transport and constitutively express BCRP. Additionally, the levels of inflammatory cytokines, interleukin 1 (IL1β), interleukin 10 (IL-10), and tumor necrosis factor α (TNFα) in the hosts’ peripheral blood were monitored. We find a clear correlation between tumor progress, the BCRP levels, and its transcript (mRNA_BCRP_) in the normal tissues, distant from the tumor site. The kinetics of the changes in BCRP expression correlate very well with the systemic inflammatory response in the host. To substantiate these effects on the functional and systemic level, we used model xenobiotics in vivo to determine whether tumor development is accompanied by any changes in the efficiency of xenobiotic efflux. This revealed a drastic, several-fold rise in the rate of xenobiotic transport in melanoma-bearing mice. Taking these observations together we show, for the first time, that BCRP expression on the molecular and systemic levels is correlated with solid tumor progress. We hypothesize that the increase in xenobiotic transport activity is not a tumor-specific response, but rather constitutes part of the universal defense strategy. The xenobiotic transport system is systemically mobilized in a pre-emptive manner, in readiness for the organism to detoxify quickly under strong inflammatory stress. Our findings shed new light on the origin of intrinsic MDR and have broad implications for cancer therapies, including chemotherapy and photodynamic therapy, and for the rational design of efficient anti-cancer drugs.

## 2. Materials and Methods

### 2.1. Animals and Experimental Groups

The animals used in the present study were obtained from the animal breeding facility at the Mossakowski Medical Research Centre, the Polish Academy of Sciences, Warsaw, Poland. Male DBA/2 mice, 2–4 months old were kept on a standard laboratory diet (LaboFeed B from Morawski, Kcynia, Poland) with free access to drinking water in community cages and a 12 h day/night regime. Before the experiments, the animals were quarantined and acclimatizated for two weeks. Their use for experimental purposes was approved by the 1^st^ Local Ethics Committee for Experiments on Animals at the Jagiellonian University in Cracow (permission No. 25/2009, 13/2010 and 132/2010).

The mice were divided into the following experimental groups: (1) three groups of melanoma-bearing mice with tumors reaching ~10 mm^3^, ~100 mm^3^, and ~1000 mm^3^ in size, respectively, and (2) healthy mice without tumors (control group). In total, 40 mice were used. After the animals had been anesthetized and their tissue sectioned, organ samples were collected and stored in liquid nitrogen until further analyses.

### 2.2. Tumor Model

The S91 Cloudman mouse melanoma cell line (subline I3) has been described previously [26]. The cells were grown as monolayers in the RPMI medium, supplemented with 5% fetal calf serum and antibiotics (penicillin, streptomycin), at 37 °C in a humidified atmosphere containing 5% CO_2_. Shortly before inoculation, cells were harvested, resuspended in 50 μL PBS and 0.5 ×10^6^ were administered subcutaneously into the hind thigh of DBA/2 mice. The tumors grew visibly over the course of the 10 days after inoculation, after which their volumes V were estimated by measuring their three perpendicular diameters (a, b, and c), using the formula: V = ^3^√(a×b×c). The immunohistochemical analysis was performed on tissues taken from animals with tumors at an advanced stage of growth, i.e., volumes exceeding 1000 mm^3^ and masses above 1 g.

### 2.3. Histological Analysis, Immunohistochemistry, and Image Analysis

The freshly excised tissue samples were immersed in cryoprotectant (Cryomatrix^TM^, Thermo Scientific, Waltham, MA, USA) and frozen in liquid nitrogen. The samples were cut into 4 µm-thick sections using a microtome (Leica CM 1100, Wetzlar, Germany), which were then placed on poly-L-lysine-coated microscopic glasses, fixed in cold 96% ethanol for 1 min and stored in PBS at 4 °C. The sections for hematoxylin and eosin staining (HE) were prepared according to a standard protocol. For immunohistochemical staining (IHC), the slides were put in PBS containing 0.1% Tween (PBS-T) for 5 min and the non-specific binding was blocked by incubation in 10% skimmed milk in PBS-T for 1.5 h. After rinsing in PBS-T (5 min), the endogenous avidin sites were blocked by incubation in egg white solution (10 min). After rinsing in distilled water (3 min), endogenous peroxidase activity was blocked by immersing in 3% H_2_O_2_ for 15 min. Then the slides were incubated with the BXP-53 antibody at a 1:100 dilution for 1 h at 30 °C (the intestines) or 2 h at room temperature (the kidneys and liver). Next, the slides were incubated with a biotinylated secondary antibody (biotin mouse anti-rat IgG 1/2a, BD Pharmingen, San Diego, CA, USA) at a 1:200 dilution for 1.5 h at room temperature. After the avidin–biotin–peroxidase complex had formed, the slides were incubated for a further 30 min, rinsed three times with PBS-T, and treated with 3,3V-diaminobenzidene (DAB; Vector Laboratories, Burlingame, CA, USA) to detect the bound peroxidase. The reaction was terminated by rinsing the slides in cold tap water. After counterstaining with hematoxylin and 0.1% ammonia water, the slides were mounted using a water-based adhesive. The negative controls were prepared by omitting the primary and secondary antibodies.

The slides were examined under a light microscope (Nikon Eclipse TS100) using a 100× magnification. The images were taken using a Nikon D7000 camera and the software Camera Control Pro 2 (Nikon Instruments, Badhoevedorp, The Netherlands). A quantitative image analysis was performed using the freeware ImageJ v1.46r (Center for Information Technology National Institutes of Health, Bethesda, MD, USA, http://rsb.info.nih.gov.ij, accessed on 10 October 2018). To estimate the percentage of the area occupied by BCRP the RGB images were transformed into 8-bit images using self-composed macros, then a binarization mask was applied, and the fractions of the area occupied by the protein were calculated. The macros included the following operations: contrast enhancement, change of brightness, binarization, and filling cavities (by erosion, dilatation, closing, and smoothing).

### 2.4. Western Blotting and Real-Time PCR

The excised tissues and organs were weighted and stored in liquid nitrogen. The samples were dissected using a scalpel, suspended in 500 µL of an ice-cold lysis buffer 1 (50 mM Tris-HCl pH 7.5, 150 mM NaCl, 1% Brij 58, 1 mM sodium orthovanadate, 1 mM PMSF, and protease inhibitors without EDTA (Roche, Basel, Switzerland)). The tissue samples were homogenized using a TH-02 tissue homogenizer (OMNI International, Tulsa, OK, USA), the homogenates were centrifuged at 16,000× *g* for 25 min at 4 °C, and the supernatants containing soluble proteins were collected and stored at −20 °C. The pellets were suspended in 400 µL of lysis buffer 2 (lysis buffer 1 containing 0.5% SDS and 1% 2-mercaptoethanol), sonicated 5 times for 15 s on ice and centrifuged at 16,000× *g* for 10 min at 4 °C. The supernatants were collected and stored at −20 °C for further analysis. The total protein concentration in the homogenates was estimated using the Bradford method. Aliquots of 5 µg of homogenates were taken for protein separation using SDS-PAGE on 5–10% polyacrylamide gels and transferred onto nitrocellulose membranes using the wet-tank electrotransfer technique (60 V, 90 min). The resulting blots were blocked in 5% non-fat milk in Tris-buffered saline containing 0.5% Tween 20 and then incubated overnight at 4 °C with the primary BXP-53 antibodies at 1:200 dilution. Mouse monoclonal anti-β-actin antibodies (Sigma-Aldrich, Hamburg, Germany) at 1:2000 dilution were used as a reference protein to maintain equal loads. After the incubation with primary antibodies, the blots were rinsed 3 times for 10 min with a Tris-buffered saline containing 0.5% Tween 20, probed with the secondary HRP-conjugated antibodies, polyclonal goat anti-rat against primary BCRP antibodies (1:2500 dilution; Enzo Life Sciences, Farmingdale, NY, USA), and polyclonal goat anti-mouse against primary β-actin antibodies (1:1000 dilution; Dako, Glostrup, Denmark) for 90 min, and rinsed again 3 times for 10 min in Tris-buffered saline containing 0.5% Tween 20. After a 1 min incubation with LumiGLO Peroxidase Chemiluminescent Reagent (Cell Signaling Technology, Danvers, MA, USA), the blots were visualized on a film in a darkroom (2 min exposure time). All the steps were carried out at room temperature. A semi-quantitative densitometric analysis of the immunoblots was performed using ImageJ.

To isolate total RNA, 30–50 mg pieces of frozen tissues were suspended in 1 mL of Qiazol (Qiagen, Hilden, Germany) and homogenized (12 min, 25 oscillations/min) using a Tissue Lyser (Qiagen, Hilden, Germany). The samples were shaken with 200 µL of chloroform (Avantor, Gliwice, Poland), incubated for 15 min on ice, and centrifuged at 12,000× *g* for 15 min at 4 °C. The upper phases were transferred into fresh test-tubes and mixed with equal volumes of cold isopropanol (Sigma-Aldrich, Schnelldorf, Germany). RNA was precipitated overnight at −20 °C, the samples were centrifuged (10,000× *g*, 30 min, 4 °C) and the resulting RNA pellets were rinsed with ice cold 70% ethanol (Avantor, Gliwice, Poland). After centrifugation, the pellets were dried at 37 °C in open test-tubes and suspended in RNAs-free water (EURx, Gdańsk, Poland). The concentration of the isolated total RNA was determined using a Nano Drop Lite spectrophotometer (Thermo Scientific, Waltham, MA, USA). The total RNA (1–3 µg) was reverse transcribed using an AMV reverse transcriptase and an oligo(dT)_20_ primer (EURx, Gdańsk, Poland). The reverse transcription was carried out for 1 h at 50 °C and terminated by heating to 85 °C for 5 min. The cDNA obtained was diluted with RNAs-free water to a concentration of 25 ng/µL. The RT-PCR was performed in triplicate, using a TaqMan^®^Gene Expression Assay with a FAM-labeled probe (Thermo Fischer Scientific, Waltham, MA, USA) and using 4 µL of cDNA per well. The reaction was carried out on an Eco Real-Time PCR System (Illumina, San Diego, CA, USA). The reaction steps were as follows: UNG activation (2 min, 50 °C), initial denaturation (10 min, 95 °C), 40 cycles including denaturation (15 s, 95 °C), and annealing/extension (1 min, 60 °C). The number of mRNA_BCRP_ copies per 100 ng of total RNA was determined on the basis of the standard curve obtained by using a synthetic oligonucleotide as a template and applying the same reaction conditions.

### 2.5. Cytokine Levels in Serum

The blood samples were collected from the heart immediately after euthanasia. The blood plasma was separated by centrifugation at 1500× *g* for 15 min at 4 °C, transferred into cryovial tubes and stored at −80 °C until analysis. In order to determine the interleukin 1 (IL1β), interleukin 10 (IL-10), and tumor necrosis factor α (TNFα) concentration in the plasma, a multiplex cytokine kit (EMD Millipore’s MILLIPLEX^®^ MAP Mouse Cytokine/Chemokine Magnetic Bead Panel, Merck, Darmstadt, Germany) was used following the instructions provided by the manufacturer. Briefly, 25 μL aliquots of serum or cytokine standards were added to the wells of a 96-well plate and diluted with the assay buffer. The mixed antibody-coupled magnetic beads were added to each well and the plate was incubated overnight on a shaker in the dark at 4 °C. Afterwards, the plate was rinsed on a handheld magnet/plate washer and re-incubated with a detection antibody solution for 1 h at room temperature. The plate was then rinsed and incubated with Streptavidin–Phycoerythrin for 30 min, rinsed again, and the beads were resuspended on the plate with Sheath fluid. The samples were analyzed using a Luminex MAGPIX^®^ reader, and the data were acquired using the xPONENT^®^ software (Hudson, MA, USA). The cytokine concentration readouts were detected as mean fluorescence intensity and converted to pg/mL of cytokine on the basis of the MFI values from the set of standards that were analyzed simultaneously in the same assay.

### 2.6. Preparation and Administration of Model Xenobiotics

Chlide, Pheide, and Zn-Pheide were prepared from Chla, as described previously [27]. A portion of solid Chlide, at a dose of 10 mg per 1 kg body weight, was taken up in a small volume (60–100 μL) of ethanol, placed in an eppendorf tube, diluted with 2 mL PBS, and sonicated for 3 min. The resulting solution was centrifuged at 2000× *g* for 10 min at room temperature in order to remove insoluble material and the supernatant was immediately injected intraperitoneally (i.p.) into the animals.

### 2.7. Xenobiotic Level in Tissues

The analyses of xenobiotic content in various tissues (blood, heart, tumor, lungs, kidneys, liver, spleen, eyes, muscles, and intestines) were carried out 0.5 and 48 h after the i.p. administration into mice. The animals were anesthetized using sodium pentobarbital (Morbital^®^, Biowet, Puławy, Poland), with ketamine (Bioketan^®^, Biowet, Puławy, Poland) and xylazine (Sedazin^®^, Biowet, Puławy, Poland) pre-medication. Their tissue samples were excised, weighted, and then stored at –30 °C until further analysis. The pigment content in tissues was determined according to the method described previously, with some modifications [27]. Briefly, the tissue samples were homogenized for 1 min in 7 mL of ice-cold 90% aqueous acetone, using the tissue homogenizer MPW-120 (MPW Medical Instruments, Warszawa, Poland) at a speed of 10,000 rpm. The homogenate was centrifuged at 2000× *g* for 10 min at 4 °C, the supernatant was collected and the pellet was re-extracted with 90% aqueous acetone to ensure complete recovery of the pigment. The extracts were pooled and analyzed spectrofluorometrically. The samples were excited at 414 nm and the fluorescence emission spectra were recorded between 600 and 850 nm. The xenobiotic content in the tissues was estimated on the basis of a calibration curve.

### 2.8. Electronic Absorption and Emission Measurements and In Vivo Fluorescence Monitoring

The absorption spectra were measured on a Cary 400 spectrophotometer (Varian, Palo Alto, CA, USA) in 1 cm quartz cuvettes at ambient temperature. The emission spectra were measured using an LS 50 Perkin-Elmer fluorometer in a 1 cm quartz cuvette at ambient temperature. The in vivo levels of Chlide and Zn-Pheide in tumors were monitored fluorimetrically after their i.p. administration at a dose of 10 mg/kg, as described previously [28]. The pigments were excited through the shaved skin using a LED source emitting at 380 nm. The emission spectra were collected using a portable USB2000 spectrometer equipped with a QR200-7-UV-Vis Fiber Fluorescence Probe (Ocean Optics, Dunedin, FL, USA).

### 2.9. Statistical Analysis

The statistical significance of the results was estimated using unpaired Student’s *t*-test, using the Microsoft Excel data analysis tool. The differences with *p* < 0.05 were considered to be statistically significant. For more details, see the captions to the figures.

## 3. Results

### 3.1. ABCG2 Protein Level

Immunohistochemical analysis (IHC), followed by optical microscopy, was performed to visualize in situ the BCRP-rich areas in the sections of the kidney, liver, and the small and large intestine (Figure 1A) of DBA/2 mice bearing a S91 melanoma tumor at an advanced stage of growth, i.e., of a mass greater than 1 g and volume ≥1000 mm^3^. For comparison, the tissues from control mice (without tumor) have also been analyzed.

In the kidney, BCRP is mainly located in the tubular epithelial cells of the nephrons, in the liver—in the hepatocytes and in the small and large intestine—in epithelial cells of the villi. A quantitative analysis of the IHC images reveals that the tumor affects the BCRP-specific staining areas and the effect is tissue-dependent (Figure 1B). Thus, BCRP coverage decreases in the kidneys (from 67 to 58%, *p* < 0.05), but increases in the liver (from 33 to 49%, *p* < 0.001), and in the small intestine (from 42 to 51%, on the borderline of significance) and large intestine (from 39 to 52%, *p* < 0.05). In each case, the results are compared within the groups “with” and “without tumor”.

The levels of BCRP in normal organs (kidney, liver, small and large intestine, heart, lungs, muscles, and tumor tissue) from healthy and S91 melanoma tumor-bearing DBA/2 mice were examined using the Western blotting (WB) technique, with the use of an anti-BCRP antibody (BXP-53) and an anti-β-actin antibody as reference. Representative results of a densitometric analysis of the blots obtained from tissues in which BCRP was detected (the band near 72 kDa), i.e., the kidney, liver, and the small and large intestine, are shown in Figure 1C. As expected, no detectable levels of BCRP were found in the tumor (not shown). In healthy mice, the highest constitutive levels of the transporter were seen in the livers (270% of the intensity of the β-actin band) and the small intestine (~150%), and much lower in the large intestine (~40%). In tumor-bearing animals, the BCRP level markedly increases in the kidneys (up to 165% of the β-actin band) but it decreases in the livers (30–90%), whereas in their intestines it remains unaffected. Characteristically, in the kidneys in both healthy and tumor-bearing animals, an additional intense band appears at ~32 kDa (Figure 1C).

### 3.2. ABCG2 Transcript

The RT-PCR technique was used to determine the levels of mRNA_BCRP_ in the tumor tissue, the small and large intestines, the liver, and kidneys obtained from four groups of mice: control (healthy, no tumor), and bearing small (V ~10 mm^3^), medium (V ~100 mm^3^), and large (V ~1000 mm^3^) tumors.

While no detectable levels of mRNA_BCRP_ were found in the tumor tissue (not shown), they vary considerably in the control group, ranging from very low (50 copies/100 ng) in the small intestine to as high as almost 250,000 copies/100 ng in the kidneys (Figure 2). Moreover, in some individuals only a low level of mRNA_BCRP_ was detected, whereas in other mice the levels of mRNA_BCRP_ may be very high, particularly in the kidneys, which indicates a large variability in mRNA_BCRP_ between the individuals. Both intestines consistently show a very low level of the transcript, near the background. In the liver it is higher (up to 55,000 copies/100 ng), while the readings from the kidneys show a large spread, from ~1000 to ~250,000 copies/100 ng. Intriguingly, in the majority of mice large discrepancies between the mRNA_BCRP_ levels in the right and left kidneys were found.

The correlations between the individual tissue levels of mRNA_BCRP_ and the tumor stage in each animal are shown in Figure 2A. In the control intestines and livers, the mRNA_BCRP_ levels are similarly low and show only a narrow spread of the values. They are higher in the other organs, but not in all individuals. In the tumor-bearing animals the differences and variations in the mRNA_BCRP_ levels become much more pronounced. In most organs the peak values are reached when the tumor size equals 100 mm^3^ and the mRNA_BCRP_ levels tend to decline in the animals with an advanced tumor (V ~1000 mm^3^), in some cases almost reaching the background level (~10000 copies/100 ng). The highest rise (~10-fold, statistically significant) is seen in the large intestine and in the right kidney (three–four-fold), when tumors reach small to medium volumes (10–100 mm^3^). Both kidneys show a high spread of mRNA_BCRP_ levels but similar differences are seen in the control. In the liver, a low level of mRNA_BCRP_ was consistently found, up to 50,000 copies/100 ng, showing no apparent effect of the tumor.

The changes in the mean levels of mRNA_BCRP_ in the control and the healthy organs of the tumor hosts across the experimental groups are shown in Figure 2B. It confirms that in the liver the expression of BCRP is the least tumor-dependent. In all the other organs, as the tumor grows it increases considerably above the control mean, reaching the maximum in mice that have small to medium sized (10–100 mm^3^) tumors, and declining almost to the control level as the tumor advances to a large size.

### 3.3. Cytokines in Serum

The concentrations of IL-1β, TNFα, and IL-10 were assessed in the peripheral blood plasma obtained from the healthy animals (control) and animals bearing S91 melanoma tumors at different stages of growth (Figure 3). The mean levels of all three cytokines in the control blood are low, and in the tumor-bearing animals they markedly (*p* < 0.02) elevate to reach the maximum values in mice bearing moderately advanced tumors (*p* < 0.002), and then plateau in mice that have an advanced tumor (100–1000 mm^3^), associated with extensive necrosis. Initially, in the control and in the early stages of the tumor, the IL-1β levels show a significant scatter among individuals, whereas those of TNFα and IL-10 have a narrower spread. Afterwards, the inter-individual variability in the levels of all three cytokines becomes large. A similar trend can be seen in the mRNA_BCRP_ levels (see above).

### 3.4. Pharmacokinetics

In order to verify the systemic effects of the tumor on xenobiotic transport, the pharmacokinetics and efficiency of xenobiotic efflux in the melanoma tumor-bearing and healthy mice were investigated. To this end, Chlide and Zn-Pheide were used as model xenobiotics and the substrates for BCRP, and the levels of endogenous Pheide in tissues from healthy and tumor-bearing mice not treated with xenobiotics were compared.

The model xenobiotics were i.p. administrated, each at a dose of 10 mg/kg body weight, in a group of healthy mice and mice bearing large tumors. Their tissue levels were examined at 0.5 h and 48 h after the injection to capture the short- and long-term effects. In the initial period, the levels of the two compounds in tumor-bearing mice, though slightly higher in the case of Chlide, are fairly comparable, whereas in healthy mice the differences are somewhat more pronounced (Figure 4A). Much larger differences are seen after 48 h, as Chlide clears almost completely from the tissues, while the clearance of Zn-Pheide is markedly slower, and, except for the intestine, its levels are about an order of magnitude higher than that of Chlide. The i.p. administrated xenobiotics near the tumor sites were also detected in vivo in a non-invasive manner, via transdermal recording of their fluorescence emission spectra using the method described previously [28]. The emission maxima located at 674 nm and 670 nm (Figure 4C) registered 1 h after xenobiotics were injected, which correspond to the positions of the emission bands of the monomeric Chlide and Zn-Pheide, respectively [27], which suggest that spectral resolution of this technique is sufficiently high.

Concerning the endogenous Pheide, its concentration in the control tissues reaches up to 1 × 10^−7^ g/g tissue, except for the spleen and intestine, where it is 7–8 times higher (Figure 4B). In contrast, only trace amounts of Pheide were detected in most tissues of tumor-bearing mice, whereas in their intestines it exceeds the average tissue level at least 10-fold.

### 3.5. Efficiency of Xenobiotic Efflux

The sections of mice treated with model xenobiotics were studied 0.5 h after they were administered. They revealed highly pigmented (dark green) areas in the small intestines of all the Chlide-treated tumor-bearing mice (8 in total), while the large intestine and the surrounding tissues remained unpigmented. As documented in photographs shown in Figure 5, the green zones of concentrated Chlide are clearly noticeable to the eye. The pigment was located mainly in the lumen of the intestine, apparently in the unmetabolized form.

Remarkably, such zones are never seen after Zn-Pheide was administered as well as in the control mice that had no tumor and were treated either with Chlide or Zn-Pheide. A fluorometric analysis confirmed a very high concentration of Chlide in the pigmented zones—as much as 66 × 10^−7^ g pigment/g tissue—in contrast to the value of 13 × 10^−7^ g pigment/g tissue in the unpigmented areas. Similarly, in the untreated tumor-bearing mice (the reference group) the mean level of dietary-derived Pheide in the intestinal tissue is twice as high (16 × 10^−7^ g pigment/g tissue) as in the mice without a tumor (Figure 5). These values are statistically significant at the 0.01 level in the case of tumor-bearing mice and at the 0.05 level in healthy mice treated with Chlide (see the caption to Figure 5 for the details). In terms of transport efficiency estimated within 0.5 h of xenobiotics being administered, in the control animals the efflux of Chlide into the small intestine amounts to 8×10^11^ molecules/s, on the average, while in the tumor-bearing ones it reaches as much as 25×10^11^ molecules/s.

## 4. Discussion

### 4.1. Tumor Effect on the BCRP Expression in Normal Tissues of the Hosts

The direct immunohistochemical detection of BCRP shows statistically significant changes in the in situ BCRP levels in normal organs from melanoma-bearing mice, distant from the subcutaneously located tumor. The BCRP levels increase in the intestines and liver and decrease in the kidney, whereas in the control group they remain practically constant. Somewhat different trends were revealed in the WB analysis (Figure 1) based on the same antibody and which in our hands underwent thorough optimization [29]. That is to say, the BCRP content in the liver cells markedly decreases, remains unchanged in the intestines and in one of the kidneys, but increases in the other kidney of the same animal. Similar discordances between these techniques have been noted in other studies [29]. An intriguing observation was made during the WB analysis of the BCRP content of the kidneys when using another antibody, BXP-53. In all the experimental groups, the BCRP band (~70 kDa) is accompanied by an additional one at approximately 32 kDa. When the antibody specific to a different epitope of the transporter (rabbit polyclonal ab63907 anti-BCRP antibody, Abcam) was used, no such polypeptide was detected (not shown). A similar band at 32 kDa was observed in the kidneys in other studies using BXP-53 [30], or in yeast using the BXP-21 antibody [31]. The lower molecular mass and strong interaction with the BCRP-specific antibodies indicate that the 32 kDa polypeptide might be a product of the proteolytic breakdown of BCRP, accumulating either due to its high turnover in the kidney, which is in agreement with the constitutively high levels of mRNA_BCRP_, and/or a post-transcriptional regulation of BCRP in this organ [32].

The levels of constitutive mRNA_BCRP_ were highest in the kidneys, lower in the liver, and lowest in the intestines, which concurs with a previous study on mice [33]. Except the liver, they are strongly affected by the developing tumor, but the responses are heterogeneous and show noticeable individual variations. Nonetheless, there is a clear rise in the mRNA_BCRP_ levels in the small and large intestine, peaking in hosts that have small-to-moderate tumors and then declining in hosts with advanced tumors. In the kidneys, the mRNA_BCRP_ levels are even more variable, both in control and in tumor hosts. Quite unexpectedly, in practically all the individuals huge differences between their left and right kidneys were found. These differences, which are two- to five-fold in most animals, including the control, in an extreme case soared to no less than 250-fold in one of the hosts. Moreover, the transient rise in the mRNA_BCRP_ level is seen only in the right kidney. These surprising results were confirmed in several runs.

The variations and rises in the BCRP transcript levels are not directly reflected in the changes in protein level as determined via the WB and IHA techniques. To some extent inter-individual variations (see below) may be responsible for this discrepancy since the two analyses were performed on different groups of mice bearing small-to-moderate tumors, but similar inconsistencies between the protein and transcript levels of BCRP have been noted in other studies [6,34]. In addition, significant inter-individual variations have been seen, for instance, in human livers [35]. These differences in BCRP estimations may indicate that its expression is subject to strict regulation. On the other hand, one should bear in mind that WB is somewhat more invasive than IHA.

Parallel to the analysis of BCRP expression, the changes in the levels of IL-1β, TNF-α, and IL-10 cytokines in peripheral blood in the control and tumor-bearing animals were investigated. To minimize spurious correlations, we focused on the anti-inflammatory IL-10, and the pro-inflammatory cytokines TNF-α and IL-1β, which are invoked in BCRP regulation [23,36,37]. Cytokine levels in tumor-bearing animals show a clear correlation with tumor growth, increasing significantly from the earliest stage of tumor development (~10 mm^3^), reaching a maximum and/or a plateau at the stage of medium size (~100 mm^3^) and then slightly declining at the later stage. In the control animals the spread of the individual values of the cytokines level is typically relatively narrow but it widens considerably in the tumor-bearing mice. Remarkably, such a pattern resembles the scatter of the individual values of mRNA_BCRP_ copies, initially narrow, very similar to the healthy group, and widening along with the tumor growth (compare Figure 2A and Figure 3).

### 4.2. Systemic Tumor Effect on Xenobiotic Transport

To substantiate the systemic effects of tumor in an independent assay, we estimated the in vivo activity of BCRP using model xenobiotics. Both compounds, Chlide and Zn-Pheide, the analogs of dietary phototoxin Pheide, are water-soluble and their pharmacokinetics in mice are known. Most Chlide is cleared unmetabolized from the body within 24 h, in contrast to Zn-Pheide, which is retained at least four times as long [27,28]. The pigments, owing to their spectral features, can be tracked either in tissue extracts or transdermally in vivo/in situ [28], which now enabled us to obtain quantitative information about xenobiotic transport. Thus, in tumor hosts, Chlide is removed into the intestine lumen rapidly and highly efficiently, which contrasts with the low efficiency of constitutive efflux in healthy animals. Due to such a drastic (five-fold) increase in transport efficiency, the efflux of Chlide reaches a macroscopic dimension and the excreted pigment is concentrated in green-colored zones in the intestines of tumor-bearing animals (see the photograph in Figure 5). The systemic effect of tumor is also evident in the reference mice, which show a significantly elevated (two-fold) efflux of endogenous Pheide. In contrast, the administration of Zn-Pheide never resulted in the occurrence of any abnormally pigmented tissue (Figure 5), which is in line with the selectivity of the active transport of Chl-derived xenobiotics [8,10,28,38]. A recent study has revealed that the affinities of the two metallocomplexes to albumins differ because of differences in the ligand-binding properties of their central metal ions, i.e., Mg^2+^ and Zn^2+^ [39]. Zn-Pheide shows a stronger affinity to human serum albumin, which in turn affects its interactions with BCRP leading to slower pharmacokinetics. It appears, that the rate of substrate transport by BCRP is related to the substrate–albumin–BCRP interplay.

These results ultimately confirm that tumor growth is associated with the systemic enhancement of xenobiotic transport, regardless of any inter-individual variations and discordances between the protein and the transcript levels found in various tissues (see above).

The fact that oncogenic stress leads to systemic upregulation of BCRP has broad consequences and sheds new light on the origin of MDR, in particular the primary type, i.e., the drug resistance that develops in cancer before any therapeutic treatments or selective pressures are applied. In most cases, the upregulation of many proteins, soluble factors, and pathways, including the ones involved in MDR, is induced due to the interactions of tumor cells with their microenvironment, e.g., stromal cells [40]. We show that the effects of a tumor are conveyed beyond its microenvironment and seem to be further amplified as they occur systemically as early as the earlier phases of tumor development. Some long-range soluble factors seem to be involved in the triggering of transporter overexpression, both in normal and tumor tissue. This implies then that the stem cell populations, often invoked in cancer drug resistance [6], play no role in inducing this type of MDR. On the other hand, the highly increased levels of transporters’ transcripts in many organs across the body, including tumor cells/tissue, may explain the prompt development of acquired MDR in response to anti-cancer agents (or other xenobiotics) [3].

A developing tumor and its microenvironment may indirectly affect the distant tissues and organs defined as the tumor macroenvironment [3,41], e.g., by causing systemic inflammation and paraneoplastic symptoms, or altering their metabolism. On the other hand, solid tumors influence the entire body through their metabolites, some of which are known to be the substrates for BCRP [42]. This sparks a question about the mechanisms and signaling pathways involved in systemic and long-range upregulation of BCRP in a tumor host. The machinery that regulates BCRP expression is intrinsically very complex owing to the existence of alternative, cell- and tissue-specific, *ABCG2* promoters, as discussed in detail in several reviews [2,12,18]. However, the majority of the *cis-* and *trans-*regulatory elements identified on the BCRP promoter seem to respond only to local factors and mainly concern cancer tissues. Therefore, they cannot be expected to play a role in systemic and long-range effects. Among the soluble and long-range acting factors, cytokines and growth- and hypoxia-related factors appear very plausible as components of the system of BCRP regulation. Figure 1 depicts the putative signaling pathways involved in the systemic response to oncogenic stress. Tumor-induced necrosis and hypoxia, oxidative stress, pro-inflammatory factors, and other strong stimuli trigger the immune response, leading to tissue-dependent changes at the level of pro-and anti-inflammatory cytokines. The inflammatory cytokines synthesized and secreted by the tumor microenvironment, including IL-1, L-6, and TNFα, cause a systemic inflammation [43,44] and there have even been attempts to extract prognostic information from changes in their patterns in cancer patients [45]. The balance between TNF-α and IL-10 is important in maintaining homeostasis and important in the case of disturbances in immune balance [46], and some anti-inflammatory cytokines, e.g., IL-10, are described as markers of advanced disease [47,48]. IL-6, IL-1β, and TNFα have already been shown to alter the expression of ABC transporters [17,23,49,50]. Moreover, oncogenic stress is associated with the activation of inflammatory transcriptome and a subsequent release of various cytokines that are known both to inhibit the proliferation of tumor cells [51] and affect the expression of xenobiotic transporters [12,18,23,36,52]. Intriguingly, the cytostatic action of cytokines, such as IL-6 and TGF-β, has been observed only at the earlier stages of tumor development [51]. This resembles the kinetics of the effects observed in the present study, i.e., the initial rise in mRNA levels followed by their decline and a plateau or decline in cytokine levels at the stage of large tumor, both accompanied by an increase in the scatter of the values. However, the majority of in vitro studies show rather the down-regulating effects of cytokines, and hence the systemic upregulation of BCRP must involve additional factors, which synergistically, or via positive feedback, cooperate with cytokines. Such a synergy has been found between cytokines and estradiol [53,54], and the transcription factor C/EBPβ, in the case of IL-6 [51,55]. In addition, there are a number of other secreted factors implicated in the senescence, which are under the control of cytokines [51]. Alternatively, other soluble factors, entirely independent, are involved in the upregulation, and the similarities between the changes in mRNA_BCRP_ and cytokine levels are merely casual. The mechanisms involved in the upregulation of BCRP, especially in humans, require further detailed study.

There is a growing body of evidence that upregulation of BCRP (as well as other xenobiotic transporters) is associated with many other diseases [56]. We hypothesize that the increase in xenobiotic transport efficiency observed in our model is a more general phenomenon. It seems to be a self-defense mechanism mobilized not only by oncogenic stimuli. This systemic response is then not a tumor-specific response but rather part of a pre-emptive defense strategy whereby the body is ready to detoxify quickly under intense inflammatory stress, in which xenobiotic transport activity is systemically mobilized via the immune system. This notion can be placed within the broader context of chemoimmunity theory, according to which the xenobiotic transport system is an essential component of the innate defense system responsible for the active efflux of low molecular mass compounds invisible to the immune system [11].

### 4.3. Implications for Cancer Therapy and Personalized Medicine

The fact that the level of BCRP expression is a highly individual feature which is systemically affected by oncogenic/inflammatory stress has important implications for the development of personalized medicine. In many cases, the interactions and retention of drugs in the body are determined and limited by the level of expression and the activity of the drug-extruding transporters, such as BCRP [57]. The intrinsic and acquired MDRs that develop in cancer tissue and in the normal organs of the hosts result in the drugs being removed from circulation more effectively and may reduce the effectiveness of therapies. In this context, analysis confined to a single tissue may not always be sufficient. More appropriate will be the assessment and monitoring of the overall xenobiotic transport activity in the patient’s body, which may become a prerequisite for personalized therapy and prognosis—the key elements of personalized medicine. Furthermore, the early upregulation of BCRP expression due to tumor opens a possibility to use this transporter as cancer biomarker; thus, establishing a rapid, reliable, and clinically available method for estimating the activity of xenobiotic transport may be indispensable for effective cancer therapy. In the case of BCRP, a convenient solution seems to be the use of water-soluble derivatives of Chl as reporting molecules and for non-invasive transdermal monitoring of their clearance in real time, using a low-cost portable spectrophotometer equipped with an optical fiber. We have shown that the transdermally monitored level of these pigments correlates with the outcome of PDT in mice bearing a melanoma tumor [28]. This would enable, in an individual patient prior to the treatment, not only potential interferences of the drug with the xenobiotic transport system to be predicted, but also, for instance, the effectiveness of the transporter inhibition to be evaluated.

## 5. Conclusions

Elevated drug efflux, mediated by xenobiotic transporters, is among the best recognized mechanisms of MDR, in which BCRP is thought to play the major role. We have found a strong correlation between the developing tumors and the level of BCRP and its transcript in normal tissues, distant from the tumor site. Additionally, the tissue responses to the tumor show considerable individual variations. At the same time, the levels of pro-inflammatory cytokines in the peripheral blood increase, and the efflux of BCRP substrates in tumor-bearing animals is drastically enhanced. To the best of our knowledge, this is the first such observation of tumor systemic effects on xenobiotic transport and BCRP expression. We hypothesize, however, that they may not be tumor-specific responses but rather a more universal defense strategy of the body under strong inflammatory stress, in which the xenobiotic transporters, seemingly in a pre-emptive manner, are systemically mobilized in order to be ready to detoxify quickly. The cytokines seem to be involved in this systemic regulation of BCRP expression, and because transporter upregulation is seen in many tissues as early as the initial stage of tumor growth, this could be the origin of “intrinsic” MDR, more basic than other mechanisms implicated in this phenomenon [3]. The progress of the disease itself induces this primary MDR, and no drug or other selection pressure is required to induce the systemic upregulation of the xenobiotic transporters. These insights into the mechanisms responsible for MDR induction and overexpression of xenobiotic transporters may form the basis for overcoming this problematic phenomenon via a strategy more effective than targeting specific transporters.

Our findings shed new light on the biology of cancer and the complexity of cancer–host interactions that should be taken into account in the design of new generations of anti-cancer drugs and personalized medicine. Such an unexpected type of tumor–host interaction, which leads to the systemic upregulation of BCRP, and very likely of other xenobiotic transporters too, has broad implications for cancer therapies, including chemotherapy and photodynamic therapy.

## Data Availability

The data presented in this study are available on request from the corresponding author.

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
