# Peer review of "Systemic Mobilization of Breast Cancer Resistance Protein in Response to Oncogenic Stress"

_cancers, 2022, doi:10.3390/cancers14020313_

Round 1

Reviewer 1 Report

It is known that breast cancer resistance protein (BCRP) is an efflux transporter that plays an important role in multidrug resistance (MDR) to anticancer drugs. Their results indicate that the expression of BCRP and the concentrations of cytokines elevated at the early stage of the tumor. Also, they concluded that the xenobiotic transport system is systematically mobilized in a pre-emptive manner, and it may not a tumor-specific response; but rather constitute a universal defense strategy to detoxify quickly and effectively. To prove their hypothesis, the experiments are well designed and arcuately executed. It is interesting that a correlation between the tumor growth, the expression of BCRP and its transcription in normal tissues far from the tumor site. The results indicated it could be a universal response and still needs to follow-up this research to prove it. However, as a preliminary data of this area, their results are strong enough and will give very valuable insights to the research community. The study will be appreciated to develop new treatments, and I recommend this manuscript to publish in Cancers with the following minor revision.

1) line 276: Student’s s test (no need s)

2) Subsection 3.1 and 3.2, the authors use ABCG2 instead of BCRP in the title. For the reader (in particular, students who are not familiar in this field), add short explanation of ABCG2 or change to BCRP.

3) In the Figure 1 caption, what is (A) in line 304 and (C) line 306?

Author Response

1) line 276: Student’s s test (no need s)

2) Subsection 3.1 and 3.2, the authors use ABCG2 instead of BCRP in the title. For the reader (in particular, students who are not familiar in this field), add short explanation of ABCG2 or change to BCRP.

3) In the Figure 1 caption, what is (A) in line 304 and (C) line 306?

We thank the reviewer for the valuable comments. According to these remarks

  • the typing mistake is now corrected
  • the abbreviations are removed from the title and the abstract, and they are defined in the Introduction
  • the inaccuracies in the figure captions have been corrected

Reviewer 2 Report

Do you see similar results in different tumor models? With only one model, might not be convincing.

The mRNA level not matching well with protein level. Different organs have so different trends. Also, IHC and western blot show conflicted results; cytokine level not matching well with the BCRP level in organs. Last but not least, variations in the same group are too big which make it difficult to draw any conclusions.  All these concern me.

A lot data missing, but did not explain why. Like, what about BCRP level in organs like lung and heart? Also, why not show the BCRP ICH staining in normal mice, then compare to tumor mice?

Overall, I do agree that tumor burden stress might will affect the distant normal organs, but this study dose need more work to make it solid.

Author Response

We thank the reviewer for the valuable and insightful comments.

Do you see similar results in different tumor models? With only one model, might not be convincing.

We are aware of some possibility, not very likely though, that the observed effects are tumor-specific. Therefore, as explained in the manuscript, our tumor and animal model has been carefully chosen in order to obtain most convincing results. Also, the bioethical issues are to be kept in mind. Anyway, we find similar tumor burden effects in a different tumor model (lung cancer LLC) applied in  different strain of mice (competent C57BL/6J)  However, this study is not yet complete.

The mRNA level not matching well with protein level. Different organs have so different trends. Also, IHC and western blot show conflicted results; cytokine level not matching well with the BCRP level in organs. Last but not least, variations in the same group are too big which make it difficult to draw any conclusions.  All these concern me.

There are several possible origins for the differences noted in our analysis. The mRNA levels are not always expected to match exactly the protein expression, mainly because of the different control mechanisms involved in the transcription (and post-) and protein synthesis and post-translational processing. In addition, the protein for the WB analysis is isolated from a homogenized tissue sample, while the IHC analysis reflects its in situ expression. There may be another source of differences in the levels of the protein detected.  Also some group discrepancies are to be expected since there are large interindividual differences between animals (population variation) seen. Although such effects may somewhat obscure the correlations but yet the results are interpretable and the trends are clear. We briefly express our concerns in the text, when discussing the WB/IHC and RT-PCR results, respectively

A lot data missing, but did not explain why. Like, what about BCRP level in organs like lung and heart? Also, why not show the BCRP ICH staining in normal mice, then compare to tumor mice?

In our study, we have focused mainly on tissues showing BCRP expression and that are known to significantly contribute to the excretion of our model xenobiotics. After finding in a few samples a very low constitutive level of the transporter in lung and heart tissues, as expected for these organs, we have decided not to carry out the entire immunohistochemical analyses of these tissues.

Surely, the expression of BCRP in tissues from control mice (tumor-free) has been examined. It is expressed in these organs and its specific signals were detected.   In further analysis, to estimate the changes and evaluate the differences in the signal intensities, the images were processed semi-automatically and in Fig. 1 the respective quantification results are shown next to the images of the slides.

 Since this procedure of image processing was made clear enough in the manuscript, now it is described more explicitly in the text (Track changes).  

Overall, I do agree that tumor burden stress might will affect the distant normal organs, but this study dose need more work to make it solid.

We think, as the other reviewers, that the effects of tumor burden is clear and solid enough, but we agree that the specific mechanisms underlying them need further elucidation.

Reviewer 3 Report

1-Systemic mobilization of BCRP in response to oncogenic stress. I would advise against the use of abbreviations in the title or abstract.

2- The multidrug resistant phenotype and cross-resistance in tumors is associated with the simulta neous overexpression of various ABC transporters, mainly P-glycoprotein, breast cancer resistance protein (BCRP or ABCG2) and multidrug resistance protein 1. ABCG2 should appear after it has been spelt out.

3- The former type of MDR exists prior to therapeutic treatment while the latter develops in response to therapeutic pressure. Could expand on how these differ functionally and mechanistically.

4- Really interesting background on transporters in development and normal physiology.

5- But not in melanoma, why not? Please explain how this has justified performing this experiment rather than using another cancer type?

6- Locally, i.e. within single organs or tissues, the expression of BCRP is under the control of several transcriptional factors, including NF-κB, HIF-1α, PPAR-γ and Nrf2. This sounds like a mechanism that could be inflammation-related or dependentc on hypoxia?

7- Additionally, the levels of inflammatory cytokines, interleukin 1 (IL1β), interleukin 10 (IL-10) and tumor necrosis factor α (TNFα) in the hosts’ peripheral blood were monitored. How about IL-6 and TNFa?

8- We hypothesize that the increase in xenobiotic transport activity is not a tumor-specific response, but rather constitutes part of the universal defense strategy.  This is interesting since many mechnaisms can activate NF-KB, i.e., it is downstream to LPS/TLR4 activation which a gram-negative bacteria molecule. Have the authors considered other scenarios such as pathogen-induced inflammation or chronic inflammation (without cancer)?

9- Before the experiments, the animals were quarantined and acclimatizated for two weeks. Were these mice tested for any type of bacterial or other infection which could induce inflammatory background noise?

10- Shortly before inoculation, cells were harvested, resuspended in 50 μl PBS and 0.5 ×106 were administered subcutaneously into the hind thigh of DBA/2 mice. I am assuming this is inaccessible like the flank to avoid scratching etc?

11- The methods are comprehensive.

12- If your hypothesis is correct, then the increase in BRCP in normal tissue in significantly less levels than tumours, suggests a systematic change in the expression levels of the protein after the onset of cancer, although how this is controlled on a systematic level is not clear.

13- Figure 1C, what is the band at 32 that appears in some samples irrespective of the whether it is normal or tumour tissue. Why did you show a few examples of tumour tissue for different organs? Couldn't you show a average of them?Why were the readings different between left and right kidney? This does seem to be the case for mRNA levels as well.  Also, the mean RNA levels don’t seem to correlate with tumour size in 2B. Why is that? Is there a predictable pattern?

14- The tumour size however corelates with cytokines in plasms, which is probably to be expected.

15- Is there data for control mice 48hour tumour/ with treatment in figure 4? There may be a better way of plotting the data between control and tumour bearing mice, so that they are matched in one plot and easier to compare. At any rate are there statistically significant differences between the organs tested between the cohorts (and also between different compounds)? Please demonstrate this.

16- They revealed highly pigmented (dark green) areas in the small intestines of all the Chlide-treated tumor-bearing mice (8 in total), while the large intestine and the surrounding tissues remained unpigmented. What is the significance of this? What is the bottom line message from figure 5?

17- At the same time, the levels of pro-inflammatory cytokines in the peripheral blood increase, and the efflux of BCRP substrates in tumor-bearing animals is drastically enhanced. How can the presence of cancer in a certain organ mechanistically affect the BRCP in other organs?

18-We hypothesize, however, that they may not be tumor-specific responses but rather a more universal defense strategy of the body under strong inflammatory stress, in which the xenobiotic transporters, seemingly in a pre-emptive manner, are systemically mobilized in order to be ready to detoxify quickly. You could use an inflammatory model as well as cancer model, or at least something that would further support this claim.

19- The cytokines seem be involved in this systemic regulation of BCRP expression. By what mechanism?

20-It would be good to have multiple samples of the subject between normal/ early-intermediate and later stage of the disease to monitor changes to BRCP levels. Although you have tried to show that with tumour volumes and there wasn’t a straightforward correlation.

21- Similar discordances between these techniques have been noted in other studies [30]. Why were the results so discordant?

22- An intriguing observation was made during the WB analysis of the BCRP 483 content of the kidneys when using another antibody, BXP-53. In all the experimental 484 groups, the BCRP band (~70 kDa) is accompanied by an additional one at approximately 485 32 kDa. When the antibody specific to a different epitope of the transporter (rabbit poly- 486 clonal ab63907 anti-BCRP antibody, Abcam) was used, no such polypeptide was detect- 487 ed (not shown).  You could show WBs with both antibodies, if you have the data.

23- The lower molecular mass and strong interaction with the BCRP-specific antibodies indicate that the 32 kDa poly-peptide might be a product of the proteolytic breakdown of BCRP, accumulating either due to its high turnover in the kidney, which is in agreement with the constitutively high levels of mRNABCRP, and/or a post-transcriptional regulation of BCRP in this organ. Would be interesting to know what this band is, especially given that the second antibody did not detect it.

24- Some topics in the discussion sounds like results and could be moved to the relevant section in the results (but it is up to the authors if they want to do that). Such as: 

Due to such a drastic (5-fold) increase in transport efficiency, the efflux of Chlide reaches a macroscopic dimension and the excreted pigment is concentrated in green-colored zones in the intestines of tumor-bearing animals (see the photograph in Fig. 5). The systemic effect of tumor is also evident in the reference mice, which show a significantly elevated (2-fold) efflux of endogenous Pheide. In contrast, the administration of Zn-Pheide never resulted in the occurrence of any abnormally pigmented tissue (Fig. 5), which is in line with the selectivity of the active transport of Chl-derived xenobiotics.

25- The systemic effect of tumor is also evident in the reference mice, which show a significantly elevated (2-fold) efflux of endogenous Pheide. In contrast, the administration of Zn-Pheide never resulted in the occurrence of any abnormally pigmented tissue. What was the justification?

Author Response

We thank the reviewer for a thorough reading of our manuscript and for the valuable and insightful comments.

1-Systemic mobilization of BCRP in response to oncogenic stress. I would advise against the use of abbreviations in the title or abstract.

Corrected as suggested.

2- The multidrug resistant phenotype and cross-resistance in tumors is associated with the simulta neous overexpression of various ABC transporters, mainly P-glycoprotein, breast cancer resistance protein (BCRP or ABCG2) and multidrug resistance protein 1. ABCG2 should appear after it has been spelt out.

Corrected as suggested. We mention in the text that other xenobiotic transporters are quite likely to also be affected by tumor burden.

3- The former type of MDR exists prior to therapeutic treatment while the latter develops in response to therapeutic pressure. Could expand on how these differ functionally and mechanistically.

As mentioned in the text, the detailed mechanisms underlying the development of the two types of MDR are not clear, in particular in the case of primary resistance, and several hypotheses/models explaining their existence have been proposed. They are still under debate (see refs 1-7). The acquired resistance is associated with increasing insensitivity to the drugs used in therapy; may result from mutations associated with target genes or proteins, as well as from alteration in the composition of the tumor microenvironment. Both types of mechanisms can coexist with each other during tumor progression. We think that our results on the upregulation of BCRP shed some light on this issue, and especially on the origin of the primary MDR as related to oncogenic stress. Accordingly, in the Discussion we put forward a model for the development of the primary MDR.

4- Really interesting background on transporters in development and normal physiology.

We appreciate this friendly remark

5- But not in melanoma, why not? Please explain how this has justified performing this experiment rather than using another cancer type?

Melanoma has been our interest for many years already, not only as a therapeutic target but also as a model solid tumor. Its advantage for the present study stems the fact of a relatively very low level of BCRP expression. Importantly, it does not elevate along tumor growth. Therefore, in the context of BCRP regulation, such a  model cancer is very suitable, also because itself it is not affected by the factors that induce BCRP expression...

6- Locally, i.e. within single organs or tissues, the expression of BCRP is under the control of several transcriptional factors, including NF-κB, HIF-1α, PPAR-γ and Nrf2. This sounds like a mechanism that could be inflammation-related or dependentc on hypoxia?

This is exactly how we interpret our observations.

7- Additionally, the levels of inflammatory cytokines, interleukin 1 (IL1β), interleukin 10 (IL-10) and tumor necrosis factor α (TNFα) in the hosts’ peripheral blood were monitored. How about IL-6 and TNFa?

We  think that monitoring of three different cytokines reveals the effect well but certainly we agree that the detailed molecular mechanism of BCRP upregulation requires further research.

8- We hypothesize that the increase in xenobiotic transport activity is not a tumor-specific response, but rather constitutes part of the universal defense strategy.  This is interesting since many mechnaisms can activate NF-KB, i.e., it is downstream to LPS/TLR4 activation which a gram-negative bacteria molecule. Have the authors considered other scenarios such as pathogen-induced inflammation or chronic inflammation (without cancer)?

and

9- Before the experiments, the animals were quarantined and acclimatizated for two weeks. Were these mice tested for any type of bacterial or other infection which could induce inflammatory background noise?

Thank you for this comment. Animal condition is an important factor that may interfere with the oncogenic stress. Therefore, a special effort was undertaken to make sure that the experimental groups bear no non-cancerous burden. Besides, any chronic or pathogen-induced inflammation would be evident in all the groups, not only in the tumor-bearing animals. Our animal breeding facility and the procedure of monitoring the mice condition were designed such as to eliminate such a risk. In addition, the animals that we receive always have a complete set of health certificates; the breeding animal facility is SPF-compliant and so infections are practically non-existent; the mice's health is constantly monitored.  

10- Shortly before inoculation, cells were harvested, resuspended in 50 μl PBS and 0.5 ×106 were administered subcutaneously into the hind thigh of DBA/2 mice. I am assuming this is inaccessible like the flank to avoid scratching etc?

The solid tumors in this location are easy to be monitored, which also allows us for the use of non-invasive imaging methods (such as in vivo EPR measurements and the transdermal fluorometric monitoring of photosensitizer level applied in our study (see Fig. 4C)).We use this model also in other studies, because it is well established and reproducible.

11- The methods are comprehensive.

We appreciate this friendly remark.

12- If your hypothesis is correct, then the increase in BRCP in normal tissue in significantly less levels than tumours, suggests a systematic change in the expression levels of the protein after the onset of cancer, although how this is controlled on a systematic level is not clear.

This comment is not clear to us because, as discussed above, the expression of BCRP is negligible in tumor itself and thus always the BCRP overexpression in normal tissues is higher.

13- Figure 1C, what is the band at 32 that appears in some samples irrespective of the whether it is normal or tumour tissue. Why did you show a few examples of tumour tissue for different organs? Couldn't you show a average of them?Why were the readings different between left and right kidney? This does seem to be the case for mRNA levels as well.  Also, the mean RNA levels don’t seem to correlate with tumour size in 2B. Why is that? Is there a predictable pattern?

We are not showing any data from tumor tissues - all the analyses (PCR, WB) have been done on normal tissues. In all case, in all tumor-bearing animals the same type tumor, melanoma, is grows in the same site. That is the point of our study.

15- Is there data for control mice 48hour tumour/ with treatment in figure 4? There may be a better way of plotting the data between control and tumour bearing mice, so that they are matched in one plot and easier to compare. At any rate are there statistically significant differences between the organs tested between the cohorts (and also between different compounds)? Please demonstrate this.

The complete pharmacokinetics of our model substrates in both control and tumor-bearing mice were already shown and analyzed by us, including the statistics, in our initial study (ref. 28) and now there was no point to sacrifice more animals to repeat the pharmacokinetic experiment. Nevertheless, in the present study a part of this experiment has been performed at selected time points to confirm consistency with the earlier observations.

16- They revealed highly pigmented (dark green) areas in the small intestines of all the Chlide-treated tumor-bearing mice (8 in total), while the large intestine and the surrounding tissues remained unpigmented. What is the significance of this? What is the bottom line message from figure 5?

The bottom line message from this results is that in tumor-bearing animal xenobiotic transport is very quick and efficient, and the substrates are very specifically recognized by the transporter (BCRP). Furthermore, this way of detoxification is much faster than the metabolic one since the compounds are concentrated in the lumen of the intestine in the unmetabolized form (green areas).

17- At the same time, the levels of pro-inflammatory cytokines in the peripheral blood increase, and the efflux of BCRP substrates in tumor-bearing animals is drastically enhanced. How can the presence of cancer in a certain organ mechanistically affect the BRCP in other organs?

As a tumor grows, extensive areas of hemorrhagic necrosis are formed, which are immunogenic in nature, which leads to an increase in the immune response involving the cytokines IL-1b, TNF-alpha, IL-10 and an increase in their concentration in the peripheral blood, which may result in enhancement of the expression of the BCRP xenobiotic transporter in intestinal and liver cells. We include this pathway in our model presented in Scheme 1.

18-We hypothesize, however, that they may not be tumor-specific responses but rather a more universal defense strategy of the body under strong inflammatory stress, in which the xenobiotic transporters, seemingly in a pre-emptive manner, are systemically mobilized in order to be ready to detoxify quickly. You could use an inflammatory model as well as cancer model, or at least something that would further support this claim.

This is a good point, and such investigations would be very insightful. However, it requires a very thorough and cautious design because it would ask a significant sacrifice of animal lives. And, which model to chose? We hope our present results will be motivating for the research groups that are specialized in such experiments.

19- The cytokines seem be involved in this systemic regulation of BCRP expression. By what mechanism?

Pro-inflammatory cytokines interact with specific nuclear receptors that are involved in regulation of BCRP expression. Several such pathways have been found (please, see refs 13 16, 18 and 20).

20-It would be good to have multiple samples of the subject between normal/ early-intermediate and later stage of the disease to monitor changes to BRCP levels. Although you have tried to show that with tumour volumes and there wasn’t a straightforward correlation.

We believe and see that there are many possible experiments to be carried out, but the animal testing and sacrifice has its limitations, as discussed above.

21- Similar discordances between these techniques have been noted in other studies [30]. Why were the results so discordant?

There are several possible origins for the differences noted in our analysis. The mRNA levels are not always expected to match the protein expression, because of the different control mechanisms involved in the transcription (and post-) and protein synthesis and post-translational processing. In addition, the protein for the WB analysis is isolated from a homogenized tissue sample, while the IHC analysis reflects its in situ expression. There may be another source of differences in the levels of the protein detected. Also group discrepancies are to be expected since there are large interindividual differences between animals (population variation) seen. Although such effects may somewhat obscure the correlations but yet the results are interpretable and the trends are clear. We briefly express our concerns in the text, when discussing the WB/IHC and RT-PCR results

22- An intriguing observation was made during the WB analysis of the BCRP 483 content of the kidneys when using another antibody, BXP-53. In all the experimental 484 groups, the BCRP band (~70 kDa) is accompanied by an additional one at approximately 485 32 kDa. When the antibody specific to a different epitope of the transporter (rabbit poly- 486 clonal ab63907 anti-BCRP antibody, Abcam) was used, no such polypeptide was detect- 487 ed (not shown).  You could show WBs with both antibodies, if you have the data.

and

23- The lower molecular mass and strong interaction with the BCRP-specific antibodies indicate that the 32 kDa poly-peptide might be a product of the proteolytic breakdown of BCRP, accumulating either due to its high turnover in the kidney, which is in agreement with the constitutively high levels of mRNABCRP, and/or a post-transcriptional regulation of BCRP in this organ. Would be interesting to know what this band is, especially given that the second antibody did not detect it.

Thank you for an interesting remark. Certainly, it is worth further exploring. Unfortunately, we have arrived to the idea of running the WB analysis with the second antibody (ab63907) a bit late, towards the end of the entire study, in fact just in order to be able to better identify the 32 kDa band. The result of the analysis is very clear and reproducible, and we can refer to it, but the blots are of somewhat unsatisfactory quality.

24- Some topics in the discussion sounds like results and could be moved to the relevant section in the results (but it is up to the authors if they want to do that). Such as: 

Due to such a drastic (5-fold) increase in transport efficiency, the efflux of Chlide reaches a macroscopic dimension and the excreted pigment is concentrated in green-colored zones in the intestines of tumor-bearing animals (see the photograph in Fig. 5). The systemic effect of tumor is also evident in the reference mice, which show a significantly elevated (2-fold) efflux of endogenous Pheide. In contrast, the administration of Zn-Pheide never resulted in the occurrence of any abnormally pigmented tissue (Fig. 5), which is in line with the selectivity of the active transport of Chl-derived xenobiotics.

We agree that explicit references to some results are made in this section of the Discussion. We think, however, that this is a better way of showing to the reader our arguments for the systemic effect of tumor. Hence, we would prefer not to break this line of reasoning.

25- The systemic effect of tumor is also evident in the reference mice, which show a significantly elevated (2-fold) efflux of endogenous Pheide. In contrast, the administration of Zn-Pheide never resulted in the occurrence of any abnormally pigmented tissue. What was the justification?

After a series of additional studies, described in two recent articles (refs 11 and 59), we are able to better interpret these differences. The slower pharmacokinetics in the case of Z-Pheide can be explained by stronger interactions of the Zn complex with albumins, if compared with the Mg complex. Evidently, the rate of substrate transport by BCRP is related to the substrate-albumin-BCRP interplay. In the revision, such an explanation is added, along with the respective references.

Reviewer 4 Report

An interesting study exploring the concentration of breast cancer resistance protein in a mouse model, finding that levels of  this protein  and its transcript correlate with tumor growth  cytokines levels both in affected and healthy organs.

I think the paper will be eligible to be published after minor revisions.

In the statistical analysis subsection, you must state what program you did use to calculate statistical significance, its maker and location. Also, did you use a paried or an unpaired t test?

Thank You

Author Response

We thank the reviewer for the comments and positive evaluation of our manuscript.

In the statistical analysis subsection, you must state what program you did use to calculate statistical significance, its maker and location. Also, did you use a paried or an unpaired t test?

The Student’s t-test, unpaired version, was applied using a Microsoft Excel software.

Round 2

Reviewer 2 Report

Thanks for the authors detailed explanation.  

But overall, the explanations did not resolve my concerns.  For example, it is true, in some cases the IHC staining will not match exactly with western blot results, but in most cases, they will show similar trend. The authors should provide more data to support their explanations, not just provide some speculations.  Furthermore, if large interinduvidual difference exist in the results, the size of tested groups should be increased, otherwise, the data will still be concerning to me. 

I do appreciate the authors efforts, but, my opinion is that the manuscript is not suitable for this journal.

Reviewer 3 Report

The authors have addressed my comments